# Fibromodulin Modulates Chicken Skeletal Muscle Development via the Transforming Growth Factor-β Signaling Pathway

**DOI:** 10.3390/ani10091477

**Published:** 2020-08-22

**Authors:** Huadong Yin, Can Cui, Shunshun Han, Yuqi Chen, Jing Zhao, Haorong He, Diyan Li, Qing Zhu

**Affiliations:** Farm Animal Genetic Resources Exploration and Innovation Key Laboratory of Sichuan Province, Sichuan Agricultural University, Chengdu 611130, China; yinhuadong@sicau.edu.cn (H.Y.); cuican123@stu.sicau.edu.cn (C.C.); hanshunshun@stu.sicau.edu.cn (S.H.); chenyuqi@stu.sicau.edu.cn (Y.C.); zhaojing@stu.sicau.edu.cn (J.Z.); hehaorong@stu.sicau.edu.cn (H.H.); diyanli@sicau.edu.cn (D.L.)

**Keywords:** Fmod, TGF-β, differentiation, muscular atrophy, myoblast

## Abstract

**Simple Summary:**

Fibromodulin (Fmod) plays critical roles in skeletal muscle development and maintenance, but the roles of Fmod in skeletal muscle atrophy and development in chickens are unclear. Here, we demonstrate that Fmod plays important roles in the differentiation and atrophy of chicken skeletal muscle by regulating the transforming growth factor-β signaling pathway. These results suggest that Fmod plays important roles in skeletal muscle growth and development in chickens.

**Abstract:**

Fibromodulin (Fmod), which is an extracellular matrix protein, belongs to the extracellular matrix small-leucine-rich proteoglycan family. Fmod is abundantly expressed in muscles and connective tissues and is involved in biological regulation processes, including cell apoptosis, cell adhesion, and modulation of cytokine activity. Fmod is the main regulator of myostatin, which controls the development of muscle cells, but its regulatory path is unknown. Chicken models are ideal for studying embryonic skeletal muscle development; therefore, to investigate the mechanism of Fmod in muscle development, Fmod-silenced and Fmod-overexpressed chicken myoblasts were constructed. The results showed that Fmod plays a positive role in differentiation by detecting the expression of myogenic differentiation markers, immunofluorescence of MyHC protein, and myotube formation in myoblasts. Fmod regulates expression of atrophy-related genes to alleviate muscle atrophy, which was confirmed by histological analysis of breast muscles in Fmod-modulated chicks in vivo. Additionally, genes differentially expressed between Fmod knockdown and normal myoblasts were enriched in the signaling pathway of transforming growth factor β (TGF-β). Both Fmod-silenced and Fmod-overexpressed myoblasts regulated the expression of TGFBR1 and p-Smad3. Thus, Fmod can promote differentiation but not proliferation of myoblasts by regulating the TGF-β signaling pathway, which may serve a function in muscular atrophy.

## 1. Introduction

Skeletal muscle is the richest muscle tissue in adult animals, accounting for 40 to 60% of body weight. Skeletal muscle plays important roles in maintaining homeostasis, initiating movement, and supporting respiration [1]. Skeletal muscle is also the most economically valuable tissue for meat production animals, the yield and quality of which directly determines the productivity benefits of animals. The development of skeletal muscle after birth mainly results from enlargement of the muscle fibers—the number of muscle fibers generally does not increase [2]. During myogenesis in the fetal phase, different types of myoblasts, including embryonic myoblasts, fetal myoblasts, and muscle satellite cells, are formed at different stages. These myoblasts undergo proliferation, migration, and differentiation to form various types of fast and slow muscle fibers [3]. This process involves expression of numerous genes, multiple signaling pathways, and regulatory networks; thus, it is of great importance when searching for candidate genes that regulate skeletal muscle development and their potential regulatory mechanisms in human and animals.

Fibromodulin (Fmod), a 42 to 80 kDa extracellular matrix (ECM) protein, belongs to the ECM small-leucine-rich proteoglycan (SLRP) family, which contains four members: lumican, biglycan, decorin, and fibromodulin [4]. Fmod is abundantly expressed in muscles, as well as in connective tissues such as cartilage, skin, and tendons [5]. Fmod functions to promote angiogenesis in cutaneous angiogenesis-dependent diseases and in the process of wound healing [6]. A number of studies have shown that Fmod participates in many biological regulation processes, including apoptosis, cell adhesion, and modulation of cytokine activity [7,8,9]. Research has shown that the Fmod expression was significantly higher in chronic lymphocytic leukemia cells than that in normal B lymphocytes, indicating that Fmod could be a tumor-associated antigen factor in chronic lymphocytic leukemia [10]. However, although Fmod has been reported to participate in myoblast growth and development, the regulatory mechanism remains unclear.

Substantial evidence confirms a prominent role of the TGF-β pathway in muscle cell proliferation, differentiation, and migration [11]. It has been reported that Fmod interacts with TGF-β to keep this profibrotic cytokine within the ECM and modulates TGF-β locally [12]. Despite the link between Fmod and the TGF-β signaling pathway having been found, the role of Fmod in the this signaling pathway during myogenesis is still unclear. In the current study, we used chicken myoblasts to determine the mechanism of Fmod in the TGF-β signaling pathway and to provide insights into this molecular entity during muscle growth and development.

## 2. Materials and Methods

### 2.1. Ethics Statement

All animal experiments were permitted by the Experimental Animal Management Committee of Sichuan Agricultural University (authorization number 2018-0418).

### 2.2. Cell Cultures

The chicken primary myoblasts were extracted from the left breast muscle of embryonic day 11 Ross 308 broilers [13]. The primary myoblasts were cultured in the growth medium (GM), including Dulbecco’s modified Eagle’s medium (DMEM; Gibco, Grand Island, NY, USA), 0.5% penicillin/streptomycin (Sigma, St. Louis, MO, USA), and 15% fetal bovine serum (FBS; Gibco, Grand Island, NY, USA). When cells reached 90% confluence, they were transferred to differentiation medium (DM) with DMEM (Gibco), 0.5% penicillin/streptomycin (Sigma, St. Louis, MO, USA), and 2% horse serum (Gibco) to induce differentiation.

### 2.3. Fmod Knockdown and Overexpression

To silence Fmod in chicken primary myoblasts, the cells were plated in 12-well plates until they had grown to 70% confluence. Then, the cells were transfected with Fmod siRNA and control siRNA. To overexpress Fmod in chicken primary myoblasts, the cells were transfected with the pcDNA 3.1 vector for Fmod and controls using Lipofectamine 3000 (Invitrogen, Carlsbad, CA, USA). Fmod siRNA sequences (sense: GCAAGAGGGUCUUUGCCAATT; antisense: UUGGCAAAGACCCUCUUGCTT) and control siRNA sequences (sense: UUCUCCGAACGUGUCACGUTT; antisense: ACGUGACACGUUCGGAGAATT) were provided by Sangon Biotech (Shanghai, China).

### 2.4. RNA Extration and Real-Time Quantitative PCR

The cellular total RNA was isolated by the miRNeasy Mini Kit (Life Technologies, Monza, Italy). Then, the RNA was reversely transcribed via the cDNA Synthesis Kit (TAKARA, Dalian, Liaoning, China). The real-time quantitative PCR (RT-PCR) followed the previous method [14]. All PCR primers were designed in the TSINGKE Primer Design Center (Beijing, China)—the primers are displayed in Table 1. The 2^−ΔΔCt^ method was used to calculate the relative expression of genes.

### 2.5. Western Blot

Once the culture medium was removed, cells were washed with PBS after removing the culture medium, then the cells were lysed by RIPA lysis buffer (Sigma, St. Louis, MO, USA). Next, the total protein was separated using 12% SDS–polyacrylamide gel electrophoresis. The concentration of protein was evaluated using the BCA protein detection kit (BestBio, Shanghai, China). The Western blotting was performed according to the previous description [15]. The primary antibodies included the rabbit anti-MyHC (Zenbio, Chengdu, Sichuan, China), rabbit anti-MyoG (Biorbyt, Cambridge, UK), mouse anti-Atrogin-1 (Novus Biological, Abingdon, UK), rabbit anti-Fmod (Santa Cruz Biotechnology, CA, USA), and rabbit anti-GAPDH (Zenbio). The secondary antibodies included the goat anti-rabbit and mouse anti-rabbit horseradish peroxidase and goat anti-mouse HRP (horseradish peroxidase), which were all purchased from Zenbio Biology Co., Ltd. (Chengdu, China).

### 2.6. Immunofluorescence Analysis

Cells were cultured on the 24-well plates in the complete medium for incubation overnight under conditions of 5% CO_2_ at 37 °C, washed by PBS twice, then fixed with 4% paraformaldehyde for 30 min. Next, after washing with PBS again, the cells were permeabilized with 0.5% Triton X-100 for 6 min. Subsequently, the cells were washed by PBS again and the cells were incubated with the MyHC antibody at 4 °C overnight. After washing with PBS, the myoblasts were incubated with the fluorescent secondary antibody at room temperature in the dark for 1 h. Finally, the fluorescence intensity was visualized using a florescence microscope (Olympus, Melville, NY, USA). The myotubule area was analyzed using Image J software (Bethesda, MD, USA).

### 2.7. Lentiviral Intramuscular Injections

The lentivirus vectors of pLKO-Fmod, pLKO.1-control, pWPXL-Fmod, and pWPXL-control were constructed by Guangzhou RiboBio Co., Ltd. (RiboBio, Guangzhou, China). The injections of lentivirus were based on previous reports [16]. We performed an initial dose on day 1 in chicks’ legs, harvested the leg muscle tissues on day 9, then fixed the muscles in 10% formalin.

### 2.8. Histomorphological Observation

The chicken leg muscles were fixed with 4% buffered formaldehyde and routinely treated with paraffin. Each tissue section (5 μm) was placed on the glass. After staining with hematoxylin and eosin (H&E), the sections were subjected to image analysis. Histological sections were observed under an optical microscope (Olympus) (Tokyo, Japan).

### 2.9. Transcriptome Analysis

Total RNA was purified from myoblasts after having been transfected with Fmod siRNA and control siRNA following the standard TRIzol protocol. The quality and quantity of RNA were analyzed using a Nanodrop NC2000 instrument (Thermo Fisher Scientific, Waltham, MA, USA) and a bioanalyzer (Agilent 2100, Agilent Technologies, Santa Clara, CA, USA), respectively. The qualified RNA samples (RIN (RNA Integrity Number) > 7; 3 independent samples from two groups, respectively) were sent for transcriptome sequencing using an Illumina Hiseq 2000 instrument at Beijing Novogene Technology Co., Ltd. (Beijing, China).

### 2.10. Statistical Analysis

The statistical software SPSS 17.0 (SPSS Inc., Chicago, IL, USA) was employed to perform all statistical analyses. Data are presented as the least squares means ± standard error of the mean (SEM). Here, *p* < 0.05 was considered as the indicator of statistical significance.

## 3. Results

### 3.1. Fmod Expression in Differentiation of Myoblasts

The Fmod mRNA and protein expression abundances were detected in chicken myoblasts. Remarkable differences were observed after 0, 24, 48, and 72 h in differentiation medium (Figure 1a). The Fmod protein levels also increased throughout the differentiation, consistent with the change in Fmod mRNA expression and MyHC protein expression (Figure 1b).

### 3.2. Interference with Fmod Restricts Formation of Myotubes

To test the role of Fmod interference in the formation of myotubes, we first performed a knockdown assay in myoblasts. Transfection with si-Fmod had a significant effect on mRNA and protein expression of Fmod (Figure 2a,b). We found morphological differences between the Fmod-silenced group and the negative control group during differentiation of myoblasts into myotubes. We explored expression levels of other genes relevant to skeletal muscle development. The mRNA expression levels of myogenic differentiation (MyoD), myogenin (MyoG), the myosin heavy chain (MyHC), and myoglobin (Mb) were drastically reduced, as were as the protein abundance levels of MyoG and MyHC (Figure 2c,d) in the Fmod-silenced group, implying an essential function of Fmod in the differentiation process. Our immunofluorescence assay results showed that MyHC was expressed at a very low level in the Fmod-silenced group compared with controls (Figure 2e,f), which suggests that Fmod knockdown inhibited myotube fusion. These data illustrated that the reduction of Fmod obstructed fusion of myotubes.

### 3.3. Overexpression of Fmod Promotes Formation of Myotubes

To test the effects of Fmod overexpression on the formation of myotubes, we conducted an overexpression assay. First, pcDNA3.1 or an Fmod overexpression plasmid (ov-Fmod) were transfected in myoblasts. The expression levels of mRNA and Fmod protein were significantly enhanced after 48 h of transfection (Figure 3a,b). The mRNA expression levels of MyoG, MyoD, MyHC, and Mb were remarkably increased, as were protein levels of MyoG and MyHC (Figure 3c,d). The immunofluorescence analysis showed increased MyHC protein in the ov-Fmod transfection group (Figure 3e,f). These results indicated that Fmod is involved in chicken myotube formation.

### 3.4. Fmod is Involved in Skeletal Muscle Atrophy in Chickens

To test whether Fmod is involved in muscular atrophy, we investigated the effect of Fmod on the atrophy-related genes including atrogin-1 and muscle ring-finger protein 1 (MuRF-1). The data showed that Fmod knockdown increased the mRNA level of atrogin-1 and MuRF-1 in myoblasts (Figure 4a) as well as the atrogin-1 protein level (Figure 4c). Conversely, Fmod overexpression (ov-Fmod) decreased the mRNA abundances of atrogin-1 and MuRF-1 in myoblasts (Figure 4b) and decreased protein expression of atrogin-1 (Figure 4c). These results indicated that Fmod plays a role in atrophy and may restrain atrophy by regulating atrophy-related genes.

Next, we performed a histology assay to determine whether Fmod controls the skeletal muscle growth in vivo. Histology revealed that the relative weight, muscle fiber cross-sectional area, and fiber diameter of breast muscle were decreased in pLKO-Fmod (knockdown) groups compared with control groups (Figure 4e–g,l) but were increased in pWPXL-Fmod (overexpression) groups (Figure 4i–k,m). These results demonstrated that Fmod may function to regulate muscular atrophy.

### 3.5. Fmod Regulates Differentiation of Myoblasts via TGF-β Signaling Pathway

To further explore the underlying molecular mechanism of Fmod regulating the muscle development, we conducted transcriptome sequencing on myoblasts transfected with si-Ctrl or si-Fmod after 3 d of culture in DM, respectively. In the si-Fmod group, we found that many genes involved in myogenesis, including MyoG, Caveolin3 (Cav3), Troponin T1 (Tnnt1), and Tropomodulin 2 (Tmod2), were markedly reduced, whereas expression of myostatin (MSTN) was increased (Figure 5a,b). In addition, we found that the TGF-β signaling pathway was highly enriched by these myogenesis-related genes (Figure 5c), so we performed another Fmod knockdown and overexpression assay to test whether Fmod acts in the TGF-β signaling pathway. Our results showed that after transfection with si-Fmod, the protein expression levels of TGFBR1 and p-Smad3 were enhanced, and vice versa (Figure 5d). The total protein level fir Smad3 remained unchanged. These results indicated that Fmod is involved in the TGF-β/Smad3 signaling pathway.

## 4. Discussion

The role of Fmod in cancer pathogenesis has been extensively studied and it has been demonstrated to be effective in arresting tumor growth. Fmod also promotes cell migration [17]. Mondal et al. demonstrated that Fmod regulates the glioma cell migration by activating the integrin-FAK-Src-Rho GTPase-dependent signaling [18]. Lee et al. reported that Fmod is a main regulator of myostatin, which controls muscle cell development during the myogenic program [19], but its regulatory pathway remains unknown. Chicken is believed to be one of the optimal models for researching the development of embryonic skeletal muscle, because it has a similar muscle developmental anatomy to that of mammals [20]. Therefore, in this study we assessed the roles and regulation mechanisms of Fmod in chicken myoblasts.

In the current study, we found that Fmod was significantly upregulated during myoblast differentiation, suggesting that Fmod may function in muscle differentiation. To elucidate the underlying mechanism of Fmod in myogenesis, chicken myoblasts were treated with Fmod siRNA. Several genes involved in skeletal muscle formation were differentially expressed, suggesting that Fmod participates in myogenesis. However, our studies showed no significant differences in Fmod expression during myoblast proliferation.

Reductions in activity and load can cause atrophy, a type of debilitation response in skeletal muscle [21]. Substantial evidence shows that muscular weakness and diminished quality of life lead to muscle atrophy [22]. During the early stages of muscle wasting, two muscle-specific ubiquitin protein ligases, atrogin-1 and MuRF-1, were detected [23]. Studies have shown that animals lacking MuRF-1 or atrogin-1 are prone to muscular atrophy after denervation [24]. Interestingly, our experiments found that Fmod severs a function in atrophy, and may inhibit atrophy by regulating atrogin-1 and MuRF-1. It has been reported that Fmod can be used in the treatment of symptomatic endometrial atrophy [25], which needs to be explored in women for whom standard estrogen management for endometrial atrophy is not available. However, little has been reported about the role of Fmod in regulating muscle atrophy. The specific molecular function of Fmod in muscle atrophy needs further study. In our study, we found that Fmod controlled the expression of atrophy-related genes Atrogin-1 and MuRF1 in myotubes. The opposite of atrophy is hypertrophy, which is caused by an increase in protein synthesis and is characterized by an increase in muscle fiber size. Lee et al. founded that Fmod promoted myoblast differentiation by inhibiting myostatin [19]. There is a significant relationship between fibrosis and muscle atrophy, whereby the muscle lost due to atrophy is replaced by fibrotic deposition, which further impairs skeletal muscle function [20]. Moreover, Fmod has been shown to be involved in fibrillogenesis. Mormone et al. showed that hepatic fibromodulin activates hepatic stellate cells and promotes collagen I deposition, which leads to liver fibrosis in mice [21]. Therefore, we speculated that Fmod knockdown in chick muscles might cause fibrosis and eventually lead to skeletal muscle atrophy in chicks.

We investigated the molecules associated with the participation of Fmod in myogenesis. Although numerous data support a single gene function of Fmod, it is unclear how Fmod participates in signaling pathways to regulate myogenesis. TGF-β is widely regarded as a pleiotropic cytokine that has three isoforms, namely TGF-β1, TGF-β2, and TGF-β3 [26]. As a potential latent precursor molecule, TGF-β is activated by interactions between proteolytic cleavage and integrin [27]. It has been proven that TGF-β is involved in cell development, angiogenesis, immune response, and inflammation [28,29,30]; signaling disorders involving TGF-β lead to multiple pathological processes, including cancer [31]. Studies have shown that in the case of Fmod−/−, deficient migration can be relieved by adding exogenous TGF-β1, which indicates that the absence of Fmod alters the normal temporospatial pattern [32]. Zheng et al. showed that Fmod interacts with TGF-β during cutaneous repair and scar formation; wound closure can be delayed in Fmod-deficient mice, which is accompanied by an increase of TGF-β3 signaling [7]. In our experiments, silencing Fmod significantly increased the protein expression of TGFBR1, whereas overexpression of Fmod reduced the TGFBR1 protein. TGF-β has been confirmed to inhibit the development of myotubes caused by the migration and fusion of newly formed myoblasts, ensuring normal development of skeletal muscle in the embryonic phase [33]. After exogenous TGF-β1 activation of Smad3, Smad3 can bind to the bHLH region of MyoD and inhibit formation of the MyoD dimer, ultimately blocking expression of specific genes of myoblasts, thus inhibiting differentiation [34]. Our results showed that silencing or overexpression of Fmod led to changes in p-Smad3, but not the whole protein of Smad3. These results indicated that one way that Fmod functions is with TGFBR1 to activate phosphorylation of Smad3.

## 5. Conclusions

In summary, we showed in this study that Fmod can promote the differentiation but not the proliferation of chicken myoblasts by regulating the TGF-β signaling pathway. Our findings provide new insights into the regulation mechanism of Fmod in skeletal muscle growth and development.

## Figures and Tables

**Figure 1 animals-10-01477-f001:**
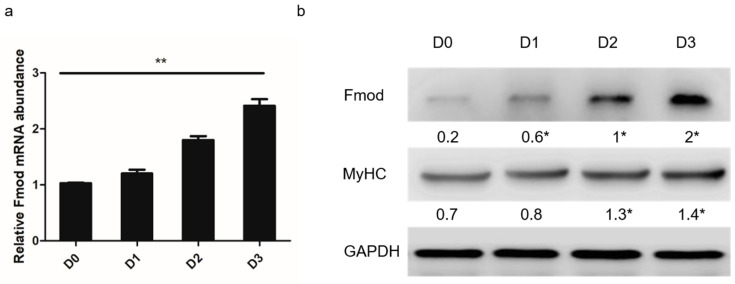
Expression of the fibromodulin (Fmod) gene in chicken primary myoblast differentiation into myotubes. (**a**) The expression levels of Fmod mRNA at 0, 24, 48, and 72 h in differentiation medium. (**b**) Western blot analysis of Fmod and MyHC protein levels at 0, 24, 48, and 72 h in differentiation medium. Note: * *p* < 0.05, ** *p* < 0.01 versus “D0”.

**Figure 2 animals-10-01477-f002:**
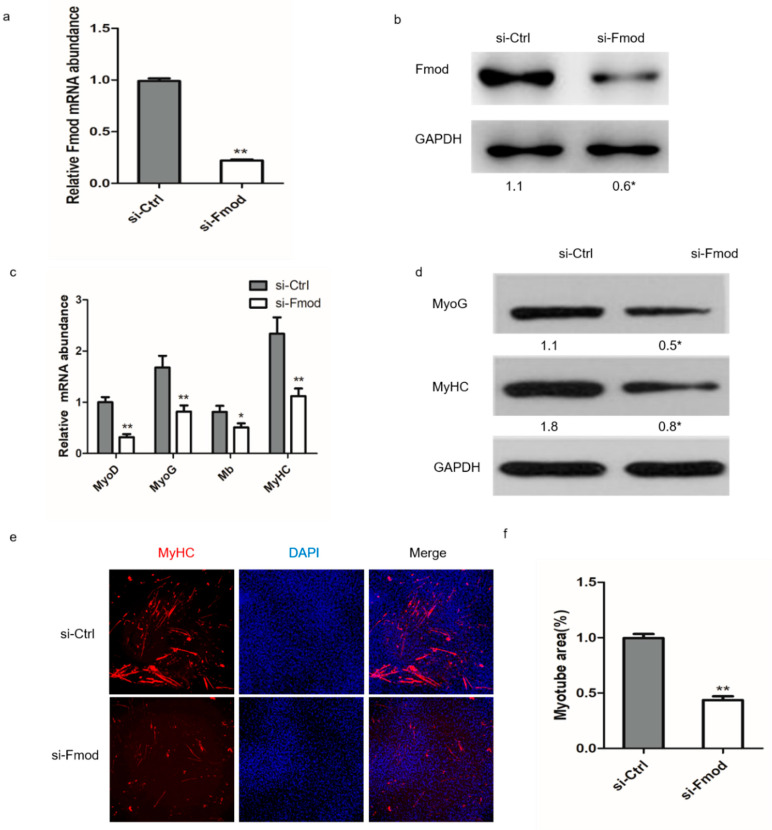
Effect of Fmod interference on chicken primary myoblast differentiation. (**a**,**b**) The Fmod mRNA and protein expression levels in myoblasts after transfection with Fmod siRNA or control siRNA for 48 h. (**c**) The MyoD, MyoG, Mb, and MyHC mRNA expression levels in myoblasts after transfection for 72 h. (**d**) Western blot analysis of MyoG and MyHC protein expression levels after transfection for 72 h. (**e**) Immunofluorescence of chicken myoblasts after transfection for 72 h. MyHc: red; DAPI: blue. (**f**) Myotube area (%) in myoblasts after transfection for 72 h. Note: * *p* < 0.05, ** *p* < 0.01.

**Figure 3 animals-10-01477-f003:**
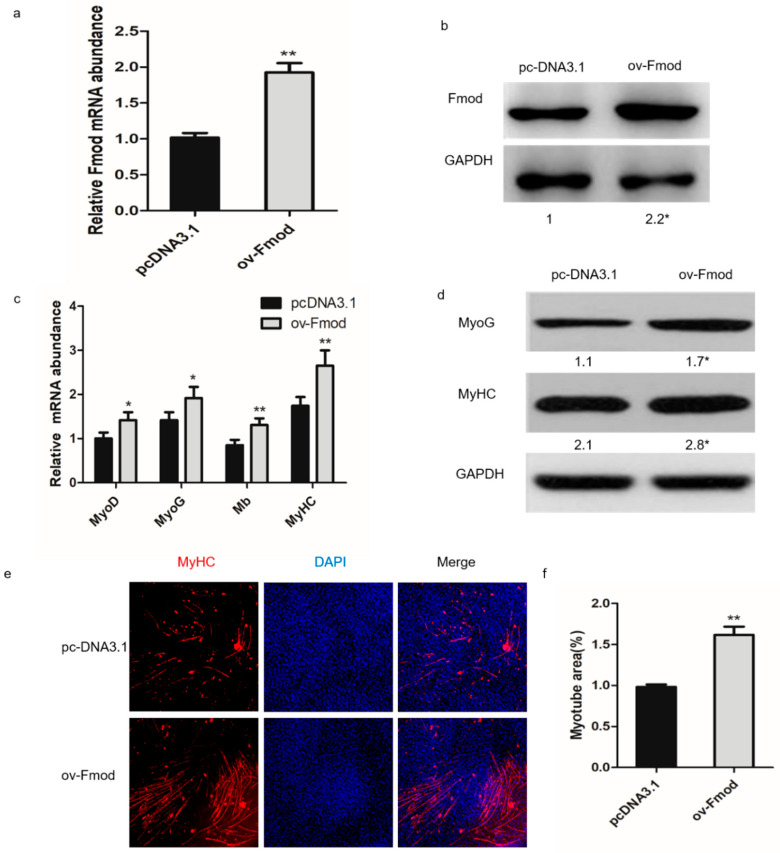
The effect of Fmod overexpression on chicken primary myoblasts differentiation. (**a**,**b**) The expression of Fmod mRNA and protein levels after transfection with pcDNA3.1 or Fmod overexpression plasmids for 48 h. (**c**) The MyoD, MyoG, Mb, and MyHC mRNA expression levels after transfection for 72 h. (**d**) Western blot analysis of MyoG and MyHC expression levels after transfection for 72 h. (**e**) Immunofluorescence of chicken myoblasts after transfection for 72 h. MyHc: red; DAPI: blue. (**f**) Myotube area (%) in myoblasts after transfection for 72 h. Note: * *p* < 0.05, ** *p* < 0.01.

**Figure 4 animals-10-01477-f004:**
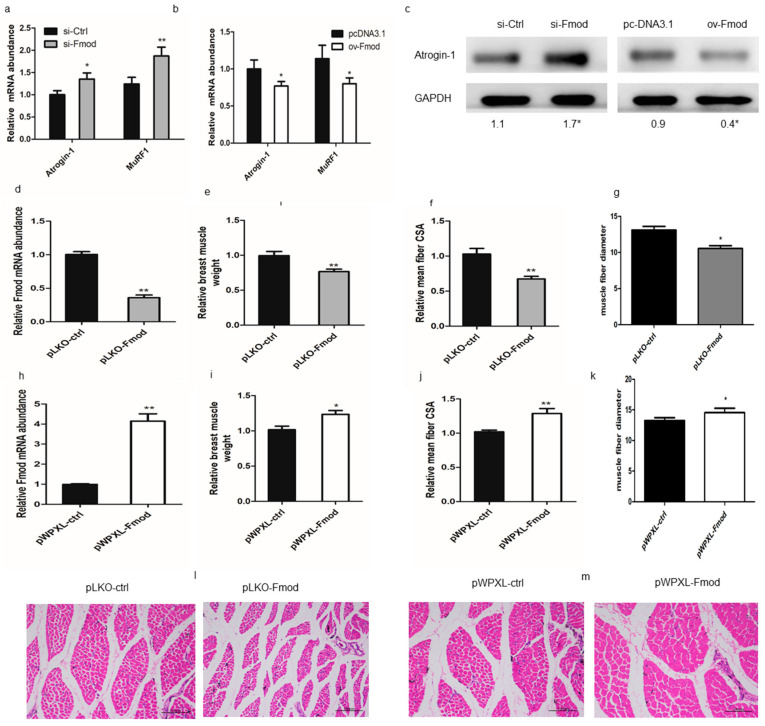
Effects of Fmod on skeletal muscle atrophy in chickens. (**a**,**b**) The expression levels of Atrogin-1 and MuRF1 after knockdown or overexpression of Fmod. (**c**) Western blot analysis of Atrogin-1 protein levels after transfection. (**d**,**h**) The Fmod mRNA expression in chick breast muscles of pLKO-Fmod and pWPXL-Fmod groups. (**e**,**i**) The relative muscle weights, (**f**,**j**) means of the muscle fiber cross-section area, and (**g**,**k**) fiber diameters of breast muscles in pLKO-Fmod and pWPXL-Fmod groups. (**l**,**m**) Hematoxylin and eosin (H&E) staining of breast muscle fiber cross-sections in pLKO-Fmod and pWPXL-Fmod groups. Note: * *p* < 0.05, ** *p* < 0.01.

**Figure 5 animals-10-01477-f005:**
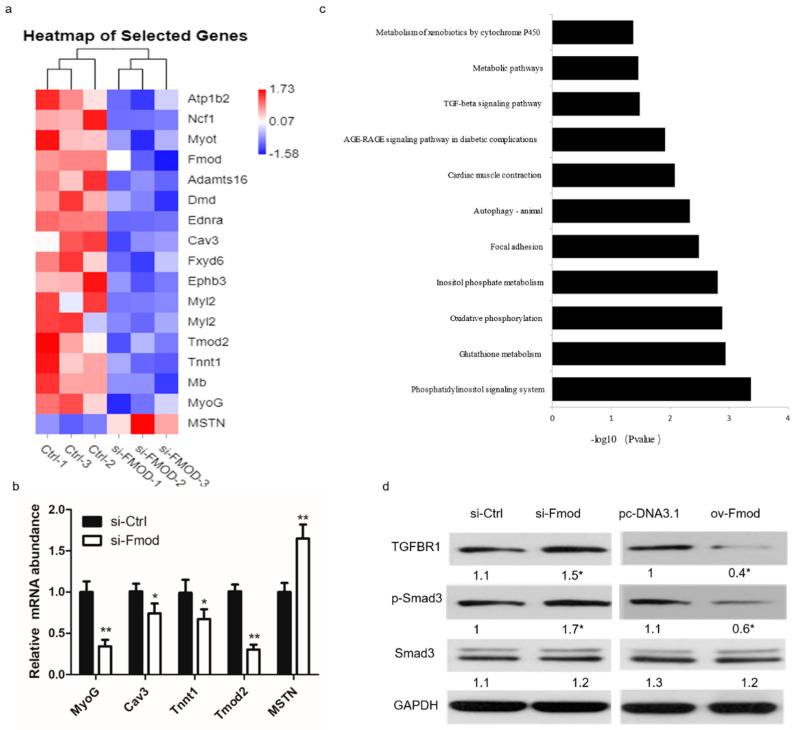
Gene expression analysis in Fmod-silenced myoblasts. (**a**) The heatmap of selected differentially expressed genes between myoblasts transfected with Fmod siRNA and control. (**b**) The mRNA expression levels of MyoG, Cav3, Tnnt1, Tmod2, and MSTN after transfection with Fmod siRNA and control. (**c**) Pathway enrichment of significant differentially expressed genes between Fmod knockdown and normal myoblasts. (**d**) The protein expression of TGFBR1, p-Smad3, Smad3, and GAPDH in myoblasts transfected with Fmod siRNA and plasmids. Note: * *p* < 0.05, ** *p* < 0.01.

**Table 1 animals-10-01477-t001:** Gene-specific primers for RT-PCR.

Gene	Forward Primer (5′-3′)	Reverse Primer (5′-3′)
Fmod	ATGGGCAGCACATCTCGATT	TCTTCGGCTTTGCAGGTCAT
MyoG	CGGAGGCTGAAGAAGGTGAA	CGGTCCTCTGCCTGGTCAT
Mb	CCCTGAGACTTTGGATCGCTT	CTGGGATTTTGTGCTTCGTGG
MyoD	GCTACTACACGGAATCACCAAAT	CTGGGCTCCACTGTCACTCA
MyHC	CTCCTCACGCTTTGGTAA	TGATAGTCGTATGGGTTGGT
Atrogin-1	TCAACGGGTCGGCAAGTCT	TCCCTCCCATCGCTCAGTC
MuRF-1	GGCAGCAGCATCATCTCGG	CCTCGCAGGTGACGCAGTAG
GAPDH	TCCTCCACCTTTGATGCG	GTGCCTGGCTCACTCCTT

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
