# Peer review of "Fibromodulin Modulates Chicken Skeletal Muscle Development via the Transforming Growth Factor-β Signaling Pathway"

_animals, 2020, doi:10.3390/ani10091477_

Round 1
Reviewer 1 Report
In this study, Yin and colleagues investigated Fibromodulin (Fmod) functions during muscle differentiation and development in chicken. The authors found Fmod promotes myoblasts differentiation. In addition, they also demonstrated Fmod inhibits atrophy-related gene expression in vitro and alleviates chicken skeletal muscle atrophy. Finally, they showed that Fmod is likely to regulate myoblast differentiation via TGF-b signaling pathway.
This report provides a new interesting perspective on the mechanisms of muscle development regulated by Fmod in Chicken. However, there are several minor points which require for publication.
1. Page3 Line91, the authors mentioned the primer sequences are displayed in Table1. But I couldn’t find Table1.
2. The authors should describe the explanation of the abbreviations, such as Mb, MyHC and MyoG.
3. In Figure 4, f and j are the same. The authors should present Relative mean fiber CSA of pLKO-Fmod in Fig. 4f.
4. The authors need to explain in M&M how did they analyze Myotube area.
Author Response
Dear reviewer:
Thank you very much for your comments concerning our manuscript entitled “Fibromodulin modulates chicken skeletal muscle development via the transforming growth factor-β signaling pathway”(ID:animals-888872). Those comments are all valuable and very helpful for revising and improving our paper, as well as the important guiding significance to our researches. We have studied comments carefully and have made correction in text, and highlighted in red color. I hope this revision can make my paper more acceptable. The main corrections in the paper and the responds to your comments are addressed point by point.
Point 1: Page3 Line91, the authors mentioned the primer sequences are displayed in Table1. But I couldn’t find Table1.
Response 1: Sorry for our carelessness, we have supplemented them in line 96.
Point 2. The authors should describe the explanation of the abbreviations, such as Mb, MyHC and MyoG.
Response 2: Thank you for your comments. We have described the full name of the abbreviations in line 155-156.
Point 3. In Figure 4, f and j are the same. The authors should present Relative mean fiber CSA of pLKO-Fmod in Fig. 4f.
Response 3: Sorry for this mistake, we have revised it in the manuscript.
Point 4. The authors need to explain in M&M how did they analyze Myotube area.
Response 4: Thank you for your suggestion. We have explained how to annlyze the myotube area in line 116.
We tried our best to improve the manuscript and made some changes in the manuscript. These changes will not influence the content and framework of the paper. We appreciate for your warm work earnestly, and hope that the response and correction will meet with approval.
Once again, thank you very much for your comments and suggestions
Reviewer 2 Report
Summary
The present study demonstrated that an extracellular matrix protein, fibromodulin (Fmod), is involved in myogenic differentiation of chicken skeletal muscle myoblasts via transforming growth factor-β (TGF-β) signal. By knockdown and overexpression of Fmod, the authors proved that Fmod promotes myotube formation in vitro and is involved in muscle development in vivo. Transcriptome analysis of the Fmod-knocked-down myoblasts revealed that Fmod is involved in TGF-β/Smad3 signaling pathway.
Almost experiments and analyses seem to be appropriately performed. To enhance the impact of this study, I hope the authors to emphasize the morphological evidence of muscle atrophy and the discussion of fibrosis.
Unfortunately, the manuscript has a lot of careless mistakes both in the text and figures as pointed below. The authors need to carefully check and revise the manuscript.
Comments
- Lines 60-61: Please show the references. “Although” should be “although” (lower case).
- Line 83: Please describe the sequences of Fmod and control siRNAs.
- Line 90: I could not find Table 1 showing primer sequences in the manuscript.
- Line 100: In which data did the authors use rabbit anti-β-actin antibody? And please describe the information of anti-GAPDH antibody used in all Western blottings.
- Line 118: Does “5xm” mean “5 μm”?
- Line 133: “Fmod function” should be “Fmod expression” as in the figure legend.
- Line 144: “FMOD Interference” should be “Fmod interference”.
- Lines 146 and 166: “FMOD” should be “Fmod”.
- Line 150: “MyoG, MyHC, and Mb” should be explained without abbreviation for the readers not familiar to muscular field.
- Line 155: “absence” should be “reduction” or some other words because Fmod protein is still existing even after si-Fmod transfection.
- Line 157: “Interference” should be “interference”.
- Line 206: “Cav3, Tnnt1, Tmod2” should be explained without abbreviation for the readers not familiar to muscular field.
- Line 240: “aMuRF-1”?
- Line 240: “Animals lacking MURF-1” should be “animals lacking MuRF-1”.
- Figures 1b, 2b, 2e, 3b, 3e, 4c, and 5d: Please quantify and statistically analyze all the results of Western blotting.
- Figures 2c and 3c: It seem to be that there is almost no difference between control and Fmod groups. Please show some quantification data (fusion index, e. g.) of the pictures. Or, I think Figures 2c and 3c can be deleted because Figures 2f and 3f showed myotube area.
- Figure 3f: MyHC and Merge images of ov-Fmod should be exchanged.
- Figure 4f and 4j are completely same. I think Figure 4f should be the fiber CSA of pLKO-Fmod.
- Figures 4d-4k: Fiber CSA demonstrate Fmod is involved in fibrosis in skeletal muscle tissue, not atrophy. In HE-staining, area or diameter of each myofiber seems to be smaller in pLKO-Fmod and to be bigger in pWPXL-Fmod. Quantification of myofiber area or diameter will be morphological evidence indicating Fmod is actually involved in atrophy.
- Figure 4d-4k: It should be that lentivirus vectors were integrated also into fibroblasts. Please discuss the relationship between the Fmod expressed in fibroblasts and the fibrosis progressed in pWPXL-Fmod.
Author Response
Dear reviewer:
Thank you very much for your comments concerning our manuscript entitled “Fibromodulin modulates chicken skeletal muscle development via the transforming growth factor-β signaling pathway”(ID:animals-888872). Those comments are all valuable and very helpful for revising and improving our paper, as well as the important guiding significance to our researches. We have studied comments carefully and have made correction in text, and highlighted in red color. I hope this revision can make my paper more acceptable. The main corrections in the paper and the responds to your comments are addressed point by point.
Point 1: Lines 60-61: Please show the references. “Although” should be “although” (lower case).
Response 1: Thank you for your suggestion. We have revised it in the manuscript.
Point 2. Line 83: Please describe the sequences of Fmod and control siRNAs.
Response 2: Thank you for your suggestion. We have described the sequences information of Fmod and control siRNAs in line 86-89.
Point 3. Line 90: I could not find Table 1 showing primer sequences in the manuscript.
Response 3: Sorry for our carelessness, we have supplemented them in line 96.
Point 4. Line 100: In which data did the authors use rabbit anti-β-actin antibody? And please describe the information of anti-GAPDH antibody used in all Western blottings.
Response 4: Sorry for this mistake. In this experiment, we used GAPDH antibody, and the GAPDH antibody information has been supplemented in line 106.
Point 5. Line 118: Does “5xm” mean “5 μm”?
Response 5: Thank you for your suggestion. We have revised it in the manuscript.
Point 6. Line 133: “Fmod function” should be “Fmod expression” as in the figure legend.
Response 6: Thank you for your suggestion. We have revised it in the manuscript in line 139.
Point 7. Line 144: “FMOD Interference” should be “Fmod interference”.
Response 7: Thank you for your suggestion. We have revised it in the manuscript.
Point 8. Lines 146 and 166: “FMOD” should be “Fmod”.
Response 8: Thank you for your suggestion. We have revised it in the manuscript.
Point 9. Line 150: “MyoG, MyHC, and Mb” should be explained without abbreviation for the readers not familiar to muscular field.
Response 9: Thank you for your suggestion. We have added the full names of these genes in line 155-156.
Point 10. Line 155: “absence” should be “reduction” or some other words because Fmod protein is still existing even after si-Fmod transfection.
Response 10: Thank you for your suggestion. We have revised it in the manuscript.
Point 11. Line 157: “Interference” should be “interference”.
Response 11: Thank you for your suggestion. We have revised it in the manuscript.
Point 12. Line 206: “Cav3, Tnnt1, Tmod2” should be explained without abbreviation for the readers not familiar to muscular field.
Response 12: Thank you for your suggestion. We have added the full names of these genes in line 208-209.
Point 13. Line 240: “aMuRF-1”?
Response 13: Thank you for your suggestion. We have revised it in line 241-242.
Point 14. Line 240: “Animals lacking MURF-1” should be “animals lacking MuRF-1”.
Response 14: Thank you for your suggestion. We have revised it in the manuscript.
Point 15. Figures 1b, 2b, 2e, 3b, 3e, 4c, and 5d: Please quantify and statistically analyze all the results of Western blotting.
Response 15: Thank you for your suggestion. We have performed quantitative statistical analysis on all Western blotting results.
Point 16. Figures 2c and 3c: It seem to be that there is almost no difference between control and Fmod groups. Please show some quantification data (fusion index, e. g.) of the pictures. Or, I think Figures 2c and 3c can be deleted because Figures 2f and 3f showed myotube area.
Response 16: Thank you for your suggestion. We have deleted Figures 2c and 3c.
Point 17. Figure 3f: MyHC and Merge images of ov-Fmod should be exchanged.
Response 17: Thank you for your suggestion. We have revised it in the manuscript.
Point 18. Figure 4f and 4j are completely same. I think Figure 4f should be the fiber CSA of pLKO-Fmod.
Response 18: Sorry for the mistake, we have revised it in the manuscript.
Point 19. Figures 4d-4k: Fiber CSA demonstrate Fmod is involved in fibrosis in skeletal muscle tissue, not atrophy. In HE-staining, area or diameter of each myofiber seems to be smaller in pLKO-Fmod and to be bigger in pWPXL-Fmod. Quantification of myofiber area or diameter will be morphological evidence indicating Fmod is actually involved in atrophy.
Response 19: Thank you for your suggestion. We have changed the 3.4 subheading ‘Fmod is involved in skeletal muscle atrophy in chickens’.
Point 20. Figure 4d-4k: It should be that lentivirus vectors were integrated also into fibroblasts. Please discuss the relationship between the Fmod expressed in fibroblasts and the fibrosis progressed in pWPXL-Fmod.
Response 20: Thank you for your suggestion. We added the sentences in line 249-255 to discuss the Fmod's relationship to muscle fiber progression.
We tried our best to improve the manuscript and made some changes in the manuscript. These changes will not influence the content and framework of the paper. We appreciate for your warm work earnestly, and hope that the response and correction will meet with approval.
Once again, thank you very much for your comments and suggestions
Round 2
Reviewer 2 Report
The authors technically revised the manuscript according to the almost reviwer’s comments except for Points 19 and 20.
In Point 19, the reviewer required the quantification of myofiber area and mentioned fibrosis in the HE-stainin images. However, the authors just changed the 3.4 subhedding.
In Point 20, the reviewer again mentioned fibrosis and fibroblasts and required to discuss them. However, the authors did not discuss it.
Fibromodulin has been studied to be involved in fibrillogenesis. I believe the discussion of fibrosis in muscle tissue is indispensable to precise understanding of muscle atrophy.
Author Response
Thanks again for your comments concerning our manuscript entitled “Fibromodulin modulates chicken skeletal muscle development via the transforming growth factor-β signaling pathway”(ID: animals-888872R1). We have studied comments carefully and have made correction, and the main revisions were highlighted by red color. I hope these revisions can meet your requirement.
Point 1: In Point 19, the reviewer required the quantification of myofiber area and mentioned fibrosis in the HE-stainin images. However, the authors just changed the 3.4 subhedding.
Response 1: Sorry for our carelessness. We have added the data of muscle fiber diameter in the figure 4g and 4k, and the related description was added in the text.
Point 2. In Point 20, the reviewer again mentioned fibrosis and fibroblasts and required to discuss them. However, the authors did not discuss it.
Fibromodulin has been studied to be involved in fibrillogenesis. I believe the discussion of fibrosis in muscle tissue is indispensable to precise understanding of muscle atrophy.
Response 2: Thank you very much for your suggestion, we have added some sentences in line 253-259 to discuss the relation of fibrosis and muscle atrophy.